# Magnitude, components and predictors of metabolic syndrome in Northern Ethiopia: Evidences from regional NCDs STEPS survey, 2016

**Kiros Fenta Ajemu**[1]*, **Abraham Aregay Desta**[1], **Asfawosen Aregay Berhe**[1], **Ataklti Gebretsadik Woldegebriel**[1], **Nega Mamo Bezabih**[1], **Degnesh Negash**[1], **Alem Desta Wuneh**[2], **Tewolde Wubayehu Woldearegay**[1]

1 Tigray Health Research Institute, Mekelle, Tigray, Ethiopia, 2 College of Health Science, Mekelle University, Mekelle, Tigray, Ethiopia

* kirosfenta@gmail.com

## Abstract

### Background

Individuals with metabolic syndrome are five times more susceptible to chronic diseases. Assessment of its magnitude, components, and risk factors is essentials to deploy visible interventions needed to avoid further complications. The study aimed to assess magnitude, components, and predictors of metabolic syndrome in Tigray region northern Ethiopia, 2016.

### Methods

Data were reviewed from Tigray region NCDs STEPs survey data base between May to June 2016. A total of 1476 adults aged 18–64 years were enrolled for the study. Multi-variable regression analysis was performed to estimate the net effect of size to risk factors associated with metabolic syndrome. Statistical significance was declared at p-value of $\leq 0.05$ at 95% confidence interval (CI) for an adjusted odds ratio (AOR).

### Results

The study revealed that unadjusted and adjusted prevalence rate of Metabolic Syndrome (MetS) were (CPR = 33.79%; 95%CI: 31.29%–36.36%) and (APR = 34.2%; 95% CI: 30.31%–38.06%) respectively. The most prevalent MetS component was low HDL concentration (CPR = 70.91%; 95%CI: 68.47%–73.27%) and (APR = 70.61; 95%CI; 67.17–74.05). While; high fasting blood glucose (CPR = 20.01% (95%CI: 18.03–22.12) and (APR = 21.72; 95%CI; 18.41–25.03) was the least ones. Eating vegetables four days a week, (AOR = 3.69, 95%CI: 1.33–10.22), a salt sauce added in the food some times (AOR = 5.06, 95%CI; 2.07–12.34), overweight (AOR = 24.28, 95%CI; 10.08–58.47] and obesity (AOR = 38.81; 12.20–111.04) had strong association with MetS.

**Data Availability Statement:** All relevant data are within the paper and its S1 File and S1 Dataset.

**Funding:** This work was not supported by any organization.

**Competing interests:** The authors declare that they have no competing interests.

**Abbreviations:** HDL, high-density lipoprotein; IDF, International Diabetes Federation; MetS, metabolic syndrome; NCEP/ATP, National Cholesterol Education Adult Treatment Panel; WHO, World Health Organization; APR, Adjusted Prevalence Rate; CPR, Crude Prevalence Rate; AOR, adjusted Odds Ratio; UK, United Kingdom; STEPs, STEP wise Approach to Surveillance.

## Conclusion

The magnitude of metabolic syndrome was found to be close to the national estimate. Community awareness on life style modification based on identified MetS components and risk factors is needed to avoid further complications.

## Introduction

Metabolic syndrome (MetS) has received much attention in recent times. It is a cluster of cardiovascular risk factors in the tune of abdominal obesity, hyperglycemia, dyslipidemia and high blood pressure [1–3]. Globally, adult population having MetS ranged from 20–25%. Individuals with metabolic syndrome are five times more susceptible to chronic diseases [3–6] and becoming an important cause of morbidity and mortality in Africa. Pieces of evidences suggested that contributing factors were rapid demographic transition, changing behaviors, and lifestyles [2–5]. The high MetS prevalence was observed in Africa ranged from 17–25% [7]. In particular; North—Western Nigeria (35.1%), South Africa (21.8%), Morocco (35.4%), and Cameroon (38.9%) [8–11].

However, in Sub-Saharan Africa (SSA) its prevalence will be 59% to 179% in 2030 [2–5]. Even though most of the studies on MetS were conducted in North America, Europe, and Asia [12–14], its impact was high in sub-Saharan African; like Kenya (25.6%), and Tanzania (38%) [15–17].

In Ethiopia the change in lifestyle due to the current rapid economic growth increased the burden of MetS [18] with an overall pooled prevalence of 20.3% [18]. Accordingly, evidence from adult treatment panel (ATP III) and international diabetic federation (IDF) showed prevalence of 12.5% and 17.9% [19–21].

Considering the literature gap on MetS prevalence and risk factors, the study aimed to assess the prevalence, components and predictors of metabolic syndrome. The evidence will use as a preliminary report to estimate the epidemiology of metabolic syndrome that will be used to promote health promotion and prevention activities for life style modification and action towards metabolic syndrome control and management.

## Materials and methods

### Design and setting

The study involved a community based cross-sectional study design. It was conducted in Tigray region Northern Ethiopia located 802KMs from Addis Ababa, the administrative capital city of Ethiopia [22]. Tigray region is the homeland of the Tigrayan, Irob and Kunama peoples. Tigray is also known as Region 1 according to the federal constitution. Its capital and largest city is Mekelle. Tigray norther Ethiopia is the 5th largest by area, the 5th most populous, and the 5th most densely populated of the 10 Regional States. Estimated total population is 7,070,260 [22] The region is further administratively subdivided into seven zones, namely, East, South, South East, Western, Northwestern, Central, and Mekelle which contained the smallest administrative units of 52 districts (34 rural and 18 urban). The study period was between May to June 2016.

### Exclusion criteria

Pregnant women and critically ill patients were excluded from the study.

## Sampling and sample size determination

As this study was part of previously published work [22], further details on the sampling technique, data collection procedure were described there. The study subjects were respondents with age categories between 18–64 years. Data were obtained from the 2016 regional STEPs survey data base (S1 Dataset). Sample size was calculated using single proportion formula; where Z-score Z α /2 = 1.96 at 95% confidence level (CL), a margin of error (d) = 5% (0.05) and assuming MetS prevalence of 18.9% (0.189) [19].

$$n = \frac{(1.96)^2 \times 0.189(0.811)}{D^2 = (0.05)^2}$$

$$n = 227$$

The total sample size was 250 by considering a 10% non-response rate. But we included 1476 respondents that had complete list of variables in the data base.

## Data collection

Data collection was carried out using a standardized questionnaire (S1 File) adopted from WHO (STEPS) instrument with slight modifications. Items had strong internal consistency (α = 0.925) [23]. Initially, the tool was prepared in English and translated into Tigrigna (local language).Three male and five female nurses with college degree and at least five-year clinical experience were recruited as data collectors. They were trained for five days on interview skills, the standard physical measurements following the WHO guideline, and blood test procedures using portable analysers. During the training, the eight data collectors conducted interviews, physical measurements, and blood tests to each two volunteers. The survey procedure was modified according to the feedback of data collectors and volunteers during the training. Two supervisors monitored the quality of the data collection. The time taken for each interview was 15 minutes. The study protocol was reviewed and approved by Mekelle University School of Public Health ethical review board. Permission was also received from Tigray Regional Health Bureau and respective health facilities. Similarly, data collection was conducted if and only if an informed consent was approved from study participant.

## Measurement and classification

After measuring each variable, the classification of MetS risk factors was made based on the cut-off values of diagnosis reference of National Heart, Lung, and Blood Institute (NHLBI) [21, 24–27].

## Operational definitions

**A waist circumference (WC).**   Waist circumference of 35 inches or more for women or 40 for men.

**A triglyceride level.**   Triglyceride level of 150 mg/dl or higher or being on medicine to treat high triglycerides.

**Cholesterol (HDL-C).**   Sometimes is called "good" cholesterol with a level of less than 50 mg/dl for women and less than 40 mg/dl for men.

**Blood pressure (BP).**   Blood pressure of 130/85 mmHg or higher or being on medicine to treat high blood pressure.

**Fasting blood sugar level.** Fasting blood sugar was considered normal for less than 100 mg/dL; pre-diabetic if between 100–125 mg/dL; while a fasting blood sugar level of 126 mg/dL or higher was considered as diabetes and a fasting blood sugar level of 100 mg/dL.

**A salt sauce added in the food.** Rarely (2 grams/day); sometimes (< 2 grams/day); always; (>2 grams/day).

**Physical inactivity.** < 600MET-minutes per week.

**Low fruit and vegetable consumption.** < five servings per day.

**Alcohol.** ≥4 standard drinks per day for men; ≥3 standard drinks per day for women.

**Metabolic syndrome.** Was considered if at least three metabolic syndromes are present, according to the National Cholesterol Education Program's Adult Treatment Panel III (NCE-P-ATP III).

## Data quality assurance

Data collectors were trained for two days. The survey procedure was modified according to the feedbacks provided during training. Completed questionnaires were checked daily by the supervisor and principal investigator.

## Data processing and analysis

Data were entered and processed using Epi-Info version 7.1.5 (Center for Disease Control and Prevention, USA) and analyzed using SPSS version 21.0 (SPSS Chicago, IL, USA). Descriptive data were presented in tables. The prevalence was described in terms of Crude & Adjusted Prevalence Rate (APR, CPR). The binary logistic regression model was used to see the net effect size. The net effect size was interpreted at P-value ≤ 0.05. Overall fitness of the model was evaluated for each logit function. The final model fitness was checked using Hosmer and Lemeshow test. Model was considered good and fit since p -value was more than 0.05 from the Hosmer-Lemeshow test. Interactions of variables were assessed at p-value < = 0.05 and confounding of variables were assessed by backward and forward elimination and any variable which had > 20% change of coefficient of the parameters between the reduced and full model was considered as confounder. Similarly, collinearity was checked by Variance Inflation Factor (VIF) and If VIF was greater than 10 it was considered as collinear and removed from the model.

# Result

## Socio-demographic characteristics

A total of 1476 respondents were enrolled in the study. Of these, 842 (57%) were females. Almost closer to half (42.7%) were age category between 29–39 years. Majorities (94%) were orthodox Christian (Table 1).

## Prevalence and components of metabolic syndrome

The most prevalent MetS component was low HDL concentration CPR = 70.91% (95%CI: 68.47%–73.27%) and APR = 70.61% (95%CI: 67.17%–74.05%). While; high fasting blood glucose CPR = 20.01% (95%CI: 18.03%–22.12%) and APR = 21.72% (95%CI: 18.41–25.03) was the least one (Table 2).

## Prevalence of metabolic syndrome

The CPR and APR of MetS were 33.79%; (95%CI: 31.29%–36.36%) and 34.2%; (95% CI: 30.31%–38.06%) respectively. The prevalence of four metabolic syndrome components was two times more than those with five components (Table 2).

**Table 1. Frequency distribution of socio-demographic characteristics of respondents in Northern Ethiopia (n = 1476).**

| Variables | Category | Frequency | |
|---|---|---|---|
| | | Number | Percent |
| Sex | Male | 842 | 57 |
| | Female | 634 | 43 |
| Age in years | 18–28 | 240 | 16.3 |
| | 29–39 | 630 | 42.7 |
| | 40–50 | 416 | 28.2 |
| | 51–61 | 162 | 11 |
| | 62+ | 228 | 1.8 |
| Marital status | Currently married | 366 | 24.8 |
| | Never married | 962 | 65.2 |
| | Separated/divorced/widowed | 148 | 10 |
| Education level | ≤8 Grade level | 101 | 6.8 |
| | 9–12 Grade Level | 175 | 11.9 |
| | >12 Grade level | 1200 | 81.3 |
| Religion | Orthodox | 1387 | 94 |
| | Muslim | 50 | 3.4 |
| | Protestant | 39 | 2.6 |

## Predictors associated to metabolic syndrome

In the adjusted analysis, currently married (AOR = 1.50; 95%CI: 1.03, 2.19), frequency of alcoholic drink 1–2 day per week (AOR = 0.60; 95%CI: 1.09, 0.71), frequency of alcoholic drink 1–3 days per month (AOR = 0.27; 95%CI:0.1, 0.76), frequency of alcoholic drink less than one per month AOR = 0.28; 95%CI: 0.11, 0.77), eating vegetables four days a week (AOR = 3.69; 95%CI:1.33, 10.22), A salt sauce added in the food sometimes (AOR = 5.06; 95%CI: 5.06 (2.07, 11.34)], heart rate average (AOR = 1.02; 95%CI: 1.01, 1.03), Hemoglobin A1C (AOR = 1.83; 95%CI: 1.49, 2.24), overweight (AOR = 24.28; 95%CI: 10.08, 58.47) and obesity = AOR = 38.81; 95%CI: 12.20, 111.04) showed statistical associations with the MetS (Table 3).

**Table 2. Prevalence of metabolic components and syndromes in Northern Ethiopia (n = 1476].**

| Metabolic syndromes and components | Metabolic syndrome | | |
|---|---|---|---|
| | Frequency | Prevalence[95% CI] | Prevalence[95%CI] |
| | | CPR | APR |
| Large waist circumference | 317 | 21.94[19.85, 24.15] | 22.60[19.23–26.00] |
| Low HDL concentration | 1046 | 70.91[68.47, 73.27] | 70.61[67.17, 74.05] |
| High Tri glyceride concentration | 842 | 57.13[54.60, 59.64] | 55.63[51.97, 59.29] |
| High blood pressure | 520 | 35.21[32.81, 37.67] | 39.14[35.46, 42.81] |
| High fasting blood glucose | 297 | 20.01[18.03, 22.12] | 21.72[18.41, 25.03] |
| At least one metabolic risk factors | 1251 | 90.92[89.27, 92.38] | 91.67[90.06, 93.27] |
| At least two metabolic risk factors | 885 | 64.32[61.72, 66.85] | 65.13[61.92, 68.34] |
| At least three metabolic risk factors | 465 | 33.79[31.29, 36.36] | 34.18[30.31, 38.06] |
| At least four risk factors | 174 | 12.65[10.93, 14.52] | 13.63[10.71, 16.55] |
| All five metabolic risk factors | 30 | 2.18[1.48, 3.10] | 3.68[1.32, 6.04] |

Note: Abbreviations: CI, Confidence Interval; CPR, Crude Prevalence Rate; APR, Adjusted Prevalence Rate.

**Table 3. Logistic regression analysis of factors associated with pre-diabetes and diabetes for study participants in Northern Ethiopia.**

| Variables | Metabolic syndrome | | OR[95%CI] | |
|---|---|---|---|---|
| | Yes | No | COR | AOR |
| **Sex** | | | | |
| Male (Ref) | 271 | 559 | | NS |
| Female | 194 | 352 | 1.14[0.91, 1.43] | |
| **Marital status** | | | | |
| Never married (Ref) | 79 | 321 | | NS |
| Currently married | 339 | 523 | 2.63[1.99, 3.49]** | **1.50[1.03, 2.19]*** |
| Separated | 2 | 3 | 2.71[0.45, 16.49] | |
| Divorced | 30 | 48 | 2.54[1.51, 4.26]** | |
| Widowed | 15 | 16 | 3.30[1.53, 7.15]** | |
| **Occupation** | | | | |
| Government employees (Ref) | 437 | 821 | | NS |
| Farmer | 18 | 49 | 0.69[0.40, 1.20] | |
| Self employed | 4 | 11 | 0.68[0.22, 2.16] | |
| Student | 1 | 17 | 0.11[0.01, 0.83]* | |
| Home marker | 13 | 15 | 1.13[0.27, 4.74] | |
| **Past smoking history** | | | | |
| Yes (Ref) | 26 | 30 | | NS |
| No | 436 | 875 | 0.57[0.34, 0.98]* | |
| **Frequency of alcoholic drink** | | | | |
| Daily (Ref) | 24 | **5** | | NS |
| 5–6 day per week | 6 | 4 | 0.31[0.06, 1.53] | |
| 1–2 days per week | 131 | 125 | 0.22[0.08, 0.59]** | **0.26[0.09, 0.71]*** |
| 1–3 days per month | 128 | 161 | 0.17[0.06, 0.45]** | **0.27[0.1, 0.76]*** |
| Less than one per month | 136 | 351 | 0.08[0.03, 0.22]** | **0.28[0.11, 0.77]*** |
| **Frequency of days eating vegetables per week** | | | | |
| 0 days (Ref) | 87 | 226 | | |
| 2 days | 126 | 214 | 1.53[1.10, 2.13]* | |
| 4days | 18 | 16 | 2.92[1.43, 5.99]** | **3.69[1.33, 10.22]*** |
| 7days | 102 | 183 | 1.45[1.02, 2.05]* | |
| **Frequency of days eating meat** | | | | |
| Once per month (Ref) | 40 | 117 | | NS |
| 3–4 times per week | 135 | 227 | 1.74[1.15, 2.64]** | |
| 5–6 times per week | 109 | 120 | 2.66[1.71, 4.14]** | |
| **A salt sauce added in the food** | | | | |
| Always (Ref) | 396 | 850 | | |
| Some times | 22 | 16 | 2.95[1.53, 5.68]** | **5.06[2.07, 12.34]**** |
| Rarely | 10 | 7 | 3.07[1.16, 8.11]* | |
| Never | 32 | 30 | 2.29[1.37, 3.82]** | |
| **Advised to quite using tobacco** | | | | |
| Yes (Ref) | 17 | 16 | | NS |
| No | 440 | 884 | 0.47[0.23, 0.94]* | |
| **Advised to eat fruits per day** | | | | |
| Yes (Ref) | 60 | 51 | | NS |
| No | 399 | 851 | 0.40[0.27, 0.59]** | |
| **Advised to reduced fat in the diet** | | | | |
| Yes (Ref) | 61 | 46 | | NS |

(*Continued*)

**Table 3.** (Continued)

| Variables | Metabolic syndrome | | OR[95%CI] | |
|---|---|---|---|---|
| | **Yes** | **No** | **COR** | **AOR** |
| No | 399 | 857 | 0.35[0.24, 0.52]** | |
| **Advised to start more physical activity** | | | | |
| Yes (Ref) | 74 | 59 | | NS |
| No | 386 | 844 | 0.36[0.25, 0.52]** | |
| **Advised to maintain weight lose** | | | | |
| Yes (Ref) | 51 | 31 | | |
| No | 48 | 871 | 0.28[0.18, 0.45]** | |
| **Heart rate average** | - | - | 1.03[1.01, 1.04]** | **1.02[1.007, 1.034]*** |
| **Hemoglobin A1C** | - | - | 2.12[1.78, 2.51]** | **1.83[1.49, 2.24]*** |
| **BMI** | | | | |
| Normal (Ref) | 218 | 571 | | |
| Under weight | 10 | 187 | 7.14[3.71, 13.74]** | **7.63[3.29, 17.73]*** |
| Overweight | 196 | 136 | 26.95[13.75, 56.97]** | **24.28[10.08, 58.47]*** |
| Obesity | 38 | 15 | 47.37[19.79, 113.40]** | **38.81[12.20, 111.04]*** |

Note: *p-value<0.05,

** P-value <0.01; Abbreviations: Ref, Reference category; NS, Not statistically significant variable.

## Discussion

The study aimed to assess the magnitude, components and predictors of MetS. It revealed that unadjusted and adjusted prevalence rate were (CPR = 33.79%; 95%CI: 31.29%–36.36%) and (APR = 34.2%; 95% CI: 30.31%–38.06%) respectively. The most prevalent component was low HDL concentration (CPR = 70.91%; 95% CI: 68.47–73.27 and APR = 70.61% 95%CI: 67.17%–74.05%). While, high fasting blood glucose was the least prevalent (CPR = 20.01%; 95%CI: 18.03%–22.12% and APR = 21.72% 95%CI: 18.41%–25.03%). eating vegetables four days a week, a salt sauce added in the food sometimes, overweight, obesity had a strong association with MetS.

Unadjusted and adjusted prevalence rate of metabolic syndrome were (CPR = 33.79%; 95% CI: 31.29%–36.36%) and (APR = 34.2%; 95% CI: 30.31%–38.06%) respectively. Several studies reported the prevalence of MetS worldwide including Africa. These shreds of evidences were quite heterogeneous, which can be attributed to a difference definitions and ways of diagnosis and classification. Hence, this limits direct comparisons with the current study. The adjusted prevalence of the current study (34.2%) lies between finding (12–86%) evidenced from sub Saharan Africa [7]. But, it is higher compared to evidence documented in 10 European countries (24%), the UK (32%) [28]. this variation was due to differences in study settings, socio-cultural, and life style modification among the countries. This prevalence is much lower than a report at Ogbera Lagos, Nigeria (86%) [29], national estimate in Ethiopia (45.9%) [19], Ghana (68.6%) [30], and Iran (64.9%) [31]. These differences could be due to the variation in diagnosis and classification criteria of MetS. The other reason for elevated value of MetS observed in Nigeria and Ghana could be due to the fast urbanization and development. Regarding to residency, the association of MetS and urbanization could be as a result of a sedentary life style, increased intake of calorie rich foods and central obesity. This result is supported by other studies [32, 33]. However, the current finding was consistent with findings from Malaysia (37.4%) [34], Germany (33.7%) [35], Korea (36.1%) [36]. These similarities might be due to the use of the same study design, definition and classification criteria.

The most prevalent component of MetS was low HDL concentration (CPR = 70.91%; 95%CI: 68.47%–73.27%) and (APR = 70.61%; 95%CI: 67.17%–74.05%) followed by high tri-glyceride concentration (CPR = 57.13%; 95%CI: 54.60%–59.64%) and (APR = 55.63%; 95%CI: 51.97%–59.29%). Similar to the current study, low HDL-C was found to be the most prevalent MetS components in black African which close (70.1%) to the current finding [37].

According to the NCEP- ATPIII criteria, where the highest prevalence of MetS was observed, high TG and low HDL were the most frequent abnormal MetS components. Even though there were no previous studies which support or contradict the findings of the current study, there were pieces of evidences that indicated abnormal levels of TG and HDL. This has an implication of adverse health effects in which low level of HDL in the body is associated with an increased risk of CVD, coronary heart diseases and death [38]. Thus, interventions focusing on abnormal TG and HDL need to be prioritized [38, 39]. Besides; a pooled prevalence (44%) of high blood pressure reviewed in Africa [7] was relatively high when compared from the current finding (39.4%). However, close to the continental estimate. The difference might be variation in design and study population in which the later was conducted using a longitudinal study design on DM patients. Even though, comparable evidence was found on WC with the current study, evidence suggested that it had the strongest associations with health risk indicators followed by BMI [40, 41].

Respondents who were currently married were significantly associated with MetS, which is also evidenced from Ethiopia [19], Nigeria [29], and Iran [31]. Eating vegetables in a typical four days a week and a salt sauce added in the food sometimes were 3.7 and 5.1 times more protected from metabolic syndrome compared to their counterparts. This evidence was further supported with studies conducted from Ethiopia [42, 43], and Brazil [44]. Besides, the odds of MetS were 24.3 and 38.8 times higher among respondents with overweight and obesity than the normal. This was almost 8 to 9 times higher when compared with the findings from Ethiopia, Nigeria Ghana, Iran, and Malaysia [19, 29, 30, 31, 34]. The difference might be due to sample size, and study setting variation. Those who drink 1–2 days a week, 1–3 days a month, less than once a month were less likely exposed to MetS than those who drink alcohol daily. Nevertheless, physical activity and use of tobacco were tended to none predictors' of MetS as similarly reported from Southern Ethiopia [45].

The findings from the present study showed MetS is a major burden. Early identification of visible intervention and awareness creation is a great importance to reduce its occurrence and progression [46].

## Strengths and limitations

The use of digital device for measured biochemical and physical measurements might increase validity and reliability of study findings. Due to the cross—sectional nature of the study the temporal relationship between the outcome and predictor variables might not powerful. The reliance on self-reported data might lead to incorrect estimates. Non-probability convenience sampling was employed and this might have an effect on generalizability.

## Conclusion

The magnitude of MetS was lower than the national estimate but significantly high considering estimations from pocket studies. The predictors were easy to address and targeted interventions of education initiatives, dietary modifications and health screening measures needed to avoid further complications.

## Supporting information

**S1 Dataset. Dataset used for the analysis of the study.**
(SAV)

**S1 File. Data collection tool.**
(PDF)

## Acknowledgments

The authors would like to thank Tigray Regional Health Bureau, Mekelle University School of Public Health, and study team for their support and contribution to the study. The authors are also grateful to the study participants.

## Author Contributions

**Conceptualization:** Kiros Fenta Ajemu, Abraham Aregay Desta, Asfawosen Aregay Berhe, Ataklti Gebretsadik Woldegebriel, Nega Mamo Bezabih, Degnesh Negash, Tewolde Wubayehu Woldearegay.

**Data curation:** Kiros Fenta Ajemu, Abraham Aregay Desta.

**Formal analysis:** Kiros Fenta Ajemu, Abraham Aregay Desta.

**Methodology:** Kiros Fenta Ajemu.

**Validation:** Kiros Fenta Ajemu, Asfawosen Aregay Berhe, Nega Mamo Bezabih, Degnesh Negash, Alem Desta Wuneh, Tewolde Wubayehu Woldearegay.

**Visualization:** Asfawosen Aregay Berhe, Ataklti Gebretsadik Woldegebriel, Nega Mamo Bezabih, Tewolde Wubayehu Woldearegay.

**Writing – original draft:** Kiros Fenta Ajemu.

**Writing – review & editing:** Kiros Fenta Ajemu, Abraham Aregay Desta, Asfawosen Aregay Berhe, Ataklti Gebretsadik Woldegebriel, Nega Mamo Bezabih, Alem Desta Wuneh, Tewolde Wubayehu Woldearegay.

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
