## [Decision Letter · Decision Letter 0]

7 Dec 2020

PONE-D-20-16786

Magnitude, components and predictors of metabolic syndrome in Northern Ethiopia: Evidences from regional NCDs STEPS survey, 2016

PLOS ONE

Dear Dr. Ajemu,

Thank you for submitting your manuscript to PLOS ONE. After careful consideration, we feel that it has merit but does not fully meet PLOS ONE’s publication criteria as it currently stands. Therefore, we invite you to submit a revised version of the manuscript that fully addresses the points raised during the review process.

We look forward to receiving your revised manuscript.

Kind regards,

Paolo Magni

Academic Editor

PLOS ONE

Journal Requirements:

2. We note that Figure 1in your submission contains map images which may be copyrighted.

We require you to either (a) present written permission from the copyright holder to publish this figure specifically under the CC BY 4.0 license, or (b) remove the figure from your submission:

b. If you are unable to obtain permission from the original copyright holder to publish this figure under the CC BY 4.0 license or if the copyright holder’s requirements are incompatible with the CC BY 4.0 license, please either i) remove the figure or ii) supply a replacement figure that complies with the CC BY 4.0 license. Please check copyright information on all replacement figures and update the figure caption with source information. If applicable, please specify in the figure caption text when a figure is similar but not identical to the original image and is therefore for illustrative purposes only.

Additional Editor Comments:

Please fully address the reviewer's comments.

Reviewers' comments:

Reviewer's Responses to Questions

**Comments to the Author**

1. Is the manuscript technically sound, and do the data support the conclusions?

Reviewer #1: Yes

2. Has the statistical analysis been performed appropriately and rigorously? 

Reviewer #1: Yes

3. Have the authors made all data underlying the findings in their manuscript fully available?

Reviewer #1: No

4. Is the manuscript presented in an intelligible fashion and written in standard English?

Reviewer #1: Yes

5. Review Comments to the Author

Reviewer #1: The title of the article is "Magnitude, components and predictors of metabolic syndrome in Northern Ethiopia: Evidences from regional NCDs STEPS survey, 2016".

The authors conducted a community based cross sectional study. This study aimed to assess magnitude, components, and predictors of metabolic syndrome in Tigray region northern Ethiopia, 2016.

This is a quite interesting study. However, the manuscript still could be further improved after some revisions.

Specific comments:

1. In Methods section; Statistical analyses, please clarify, what method that used for adjusting in multivariate analysis? Please provide test for interaction between variables, goodness of fit, and multicollinearity. In addition, please demonstrate flowchart of participants. If non-probability convenience sampling was employed. This limitation of the study might affect generalizability.

2. Data collection: Who interviewed the participants? Were they doctors, nurses, medical students, or research investigators? Were they trained before administered the questionnaires? How the authors deal with missing data?

3. Data collection: By how many people, in how long time, where? The time of interview for each person? Missing values?

4. Include full details of how the authors handled missing data and outliers in the ‘Methods’ section.

5. The main concern is that the questionnaires should be validated and have good reliability and validity. Reliability of questionnaires should be mentioned. Please provide citation and reference of the questionnaire. The English peer-review reference should be placed.

6. Please describe the detail of Tigray region northern Ethiopia; such as is it rural or urban community, the number of population and population structure.

7. It is important that within the manuscript, the authors clarify the importance of this work, how it differs from and advances previously published work and how this article can benefit the field and patients in the future etc. Please also add more information from recently published research and offer a more speculative and forward-looking perspective.

6. PLOS authors have the option to publish the peer review history of their article (what does this mean?). If published, this will include your full peer review and any attached files.

Reviewer #1: **Yes: **Wisit Kaewput

---

## [Author Response · Author response to Decision Letter 0]

21 May 2021

Responses to Reviewer’s and editor comments 

Journal name: PLOS ONE

Corresponding Author: Kiros Fenta Ajemu

We would like to express our appreciation to the reviewer for their constructive and supportive comments in the first version of the manuscript entitled “Magnitude, components and predictors of metabolic syndrome in Northern Ethiopia: Evidences from regional NCDs STEPS survey, 2016”. This will be an input to improve the quality of the manuscript. We have meticulously revised the manuscript and incorporated all the changes in the revised version of the manuscript based on the suggestions and comments made by the reviewer. We have highlighted these changes in the manuscript. We have listed all the responses for each of the comment/ suggestion made by the reviewer as follows:

Responses to comments Academic Editor: Mr. Paolo Magni

Comment 1: Please ensure that your manuscript meets PLOS ONE's style requirements, including those for file naming.

Response 1: Thank you for your constructive comment. We crosscheck the manuscript based on the PLOS ONE style requirements in line with the web site you suggested. 

Comment 2: We noted that “Figure 1” in your submission contains map images which may be copyrighted.

Response 2: Thank you for your transparent and constructive comment. Actually we made the map using regional GPS data but as you have said the map is not that much relevant since Mekelle city is already known and everybody can Google it from internet. Therefore we agreed to remove the figure “Figure 1” from the main body of the manuscript. The change made is listed below as per your comments and suggestions in:

- Material and Methods section Line (70), Page (4)].

Responses to reviewer comments

Comment 1: The title of the article is "Magnitude, components and predictors of metabolic syndrome in Northern Ethiopia: Evidences from regional NCDs STEPS survey, 2016".The authors conducted a community based cross sectional study. This study aimed to assess magnitude, components, and predictors of metabolic syndrome in Tigray region northern Ethiopia, 2016.This is a quite interesting study. 

Response 1: thank you for your acknowledgment that makes me to do more.

Comment 2: in the statistical analyses, please clarify what method that used for adjusting in multivariate analysis? Please provide test for interaction between variables, goodness of fit, and multicollinearity.

Response 2: thank you for your consideration in statical analysis in line with data handling mechanisms since it is the back bone and pillar to find the real evidence and minimize biases and errors. The changes that we made were listed below as per your comments and suggestions in Method section, Line (137-144), Pages (7)].

- Test of interaction, Line (142-143), Pages (7)].

- Goodness of fit, Line (140-142), Pages (7)].

- Multicollinearity, Line (145-146), Pages (7)].

Comment 3: If non-probability convenience sampling was employed. This limitation of the study might affect generalizability.

Response 3: thank you and I accept you fear of generalizability but we put it as a limitation in the:

- Strength and Limitation section, Line (243-244), pages (15)]. As part of the limitation of the study

Comment 4: Who interviewed the participants? Were they doctors, nurses, medical students, or research investigators? Were they trained before administered the questionnaires, by how many people, in how long time, where? The time of interview for each person? 

Response 4: thank you your concern for data quality and the changes made were explained in Method section, Line (94- 103] Page (5)] 

- Who interviewed, Line (96-97), Pages (5)].

- How many people, Line (96-97), Pages (5)].

- How long and where, Line (102-103), Pages (5)].

Comment 5: The main concern is that the questionnaires should be validated and have good reliability and validity. Reliability of questionnaires should be mentioned Please provide citation and reference of the questionnaire. 

Response 5: thank you for your comment and the changes made were listed in Method section Line (94-95] Page (5)]

- Validated questionnaire, Line (94), Pages (5)].

- Reference, Line (95), Pages (5)].

Comment 6: Please describe the detail of Tigray region northern Ethiopia; such as is it rural or urban community, the number of population and population structure.

Response 6: thank you for your concern in the study area and the changes made were listed in Method section Line (70-77] Page (4)]

Comment 7: It is important that within the manuscript, the authors clarify the importance of this work, how it differs from and advances previously published work and how this article can benefit the field and patients in the future etc.

Response 7: the previously published work has different objective and it was a scientific report on prevalence of behavioral characteristics such as alcohol use, eating diet, physical activity, diabetes and pre-diabetes prevalence. The study did not address predictor factors the magnitude of association in respective cardiovascular disease. The current study will provide detail of the risk factors for future deployment of visible intervention. For further information you can access the article on line Journal of scientific report https://www.ncbi.nlm.nih.gov/pmc/articles/PMC6006379/.

Once again thank you for your strong and committed concern for improving the manuscript

Sincerely your’s Kiros Fenta Ajemu (kirosfenta@gmail.com)Researcher, Tigray Health Research Institute, Mekelle, Tigray, Ethiopia

---

## [Decision Letter · Decision Letter 1]

3 Jun 2021

Magnitude, components and predictors of metabolic syndrome in Northern Ethiopia: Evidences from regional NCDs STEPS survey, 2016

PONE-D-20-16786R1

Dear Dr. Kiros Fenta Ajemu,

We’re pleased to inform you that your manuscript has been judged scientifically suitable for publication and will be formally accepted for publication once it meets all outstanding technical requirements.

Kind regards,

Paolo Magni

Academic Editor

PLOS ONE

Additional Editor Comments (optional):

Reviewers' comments:

Reviewer's Responses to Questions

**Comments to the Author**

1. If the authors have adequately addressed your comments raised in a previous round of review and you feel that this manuscript is now acceptable for publication, you may indicate that here to bypass the “Comments to the Author” section, enter your conflict of interest statement in the “Confidential to Editor” section, and submit your "Accept" recommendation.

Reviewer #1: All comments have been addressed

2. Is the manuscript technically sound, and do the data support the conclusions?

Reviewer #1: Yes

3. Has the statistical analysis been performed appropriately and rigorously? 

Reviewer #1: Yes

4. Have the authors made all data underlying the findings in their manuscript fully available?

Reviewer #1: Yes

5. Is the manuscript presented in an intelligible fashion and written in standard English?

Reviewer #1: Yes

6. Review Comments to the Author

Reviewer #1: (No Response)

7. PLOS authors have the option to publish the peer review history of their article (what does this mean?). If published, this will include your full peer review and any attached files.

Reviewer #1: **Yes: **Wisit Kaewput

---

## [Editor Report · Acceptance letter]

7 Jun 2021

PONE-D-20-16786R1 

Magnitude, components and predictors of metabolic syndrome in Northern Ethiopia: Evidences from regional NCDs STEPS survey, 2016 

Dear Dr. Ajemu:

I'm pleased to inform you that your manuscript has been deemed suitable for publication in PLOS ONE. Congratulations! Your manuscript is now with our production department. 

Kind regards, 

on behalf of

Prof. Paolo Magni 

Academic Editor

PLOS ONE